# Analysis of Ionicity-Magnetism Competition in 2D-MX3 Halides towards a Low-Dimensional Materials Study Based on GPU-Enabled Computational Systems

**DOI:** 10.3390/nano11112967

**Published:** 2021-11-05

**Authors:** Alexey Kartsev, Sergey Malkovsky, Andrey Chibisov

**Affiliations:** 1Computing Center of the Far Eastern Branch of the Russian Academy of Sciences, 680000 Khabarovsk, Russia; sergey.malkovsky@ccfebras.ru (S.M.); andreichibisov@yandex.ru (A.C.); 2Pacific National University, 680035 Khabarovsk, Russia

**Keywords:** density functional theory, 2D magnets, CrI3, RuCl3, biquadratic exchange, ferromagentism, CUDA, ESSL, CPU, GPU

## Abstract

The acceleration of parallel high-throughput first-principle calculations in the context of 3D (three dimensional) periodic boundary conditions for low-dimensional systems, and particularly 2D materials, is an important issue for new material design. Where the scalability rapidly deflated due to the use of large void unit cells along with a significant number of atoms, which should mimic layered structures in the vacuum space. In this report, we explored the scalability and performance of the Quantum ESPRESSO package in the hybrid central processing unit - graphics processing unit (CPU-GPU) environment. The study carried out in the comparison to CPU-based systems for simulations of 2D magnets where significant improvement of computational speed was achieved based on the IBM ESSL SMP CUDA library. As an example of physics-related results, we have computed and discussed the ionicity-covalency and related ferro- (FM) and antiferro-magnetic (AFM) exchange competitions computed for some CrX${}_{3}$ compounds. Further, it has been demonstrated how this exchange interplay leads to high-order effects for the magnetism of the 1L-RuCl${}_{3}$ compound.

## 1. Introduction

Since the first discovery of graphene, 2D materials have been proposed as a designing basis of novel quantum systems and devices for applications in energy harvesting and microelectronics. Recently, for the first time, the monolayer 2D CrI${}_{3}$, ferromagnet with a honeycomb Cr-sublattice has been fabricated [1], followed by the cascade of experimental and theoretical studies about a new class of 2D magnets [2]. In that fashion high-throughput simulation of 2D magnets is now an emerging field in solid-state physics [2,3,4], with a growing need for new parallel computational methods based on various numerical techniques. First-principal understanding and predicting of 2D magnetic system properties from the computational point of view is challenging because of the need to capture the whole complexity of the interplay between quantum effects. Where large-scale nano-constructions, such as CrI${}_{3}$ nanotubes, show even more fascinating electronic and magnetic properties [5]. Therein marks the need for efficiency improvement of first-principle codes and their scaling over on high-performance systems.

In the last decade GPUs have been actively developing the direction of high-performance computing systems, where classical CPU-based systems reached their limits. They have shown high computational efficiency and, therefore, have been increasingly used for a simulation purpose in the various fields of physics [6,7,8], where an optimal choice for configuration of the parallel environment is the key point to reach the best performance [9]. In this work, we focus on the testing and comparison of computer design methods applied to these materials in the frame of the Quantum ESPRESSO package using both CPU and GPU enhanced versions. The main branch of GPU-accelerated Quantum ESPRESSO- pw.x module has been tested for electronic properties calculations. Computational costs and scaling tests have been performed using the hybrid GPU-CPU environment and the pure CPU environment where the 2D CrI${}_{3}$ magnets have been used as a testing system. Two types of pseudopotentials have been employed - using the ultrasoft pseudopotentials (US) approach and the projector augmented-wave method (PAW).

## 2. Hardware and Parallel Environment

The hybrid computing cluster used consists of one control and four computing nodes. They are Sitonica PW22LC servers (IBM Power System S822LC 8335-GTB). Where each node includes two IBM POWER8 processors with a maximum frequency of 4.023 GHz, two NVIDIA Tesla P100 GPU co-processors (NVIDIA Corporation, 2701 San Tomas Expressway, Santa Clara, CA, USA), 256 GB DDR4 RAM, an EDR InfiniBand controller, and two Seagate ST1000NX0313 1 TB 7200RPM hard drives (Seagate Technology, Inc. 10200 South De Anza Boulevard, Building One, Cupertino, CA, USA). In order to store data, a storage area network with 50 TB capacity was used. A management network based on Gigabit Ethernet technology was used. While for the data network, the EDR InfiniBand network with a capacity of 100 Gb/s was employed.

For comparison purposes, three different builds of Quantum ESPRESSO (QE) version 6.2 open-source package were used. The first one (CPU) is a simple CPU-version without GPU-support, in which as a mathematical library, the serial version of the IBM ESSL version 6.2.1 (Engineering and Scientific Subroutine Library) library was used. The second one (GPU) is the Quantum ESPRESSO version 6.2 with GPU-support activated with a special open-source extension QE-GPU-Plugin [10].

The serial version of the IBM ESSL version 6.2.1 library was also used as a mathematical library in this build. The third one (ESSLCUDA) is the standard CPU-version compiled along with the GPU-enabled version of the ESSL library, which supports the offloading part of the calculations into co-processors. All versions were compiled by IBM compilers (IBM XL Fortran version 15.1.5, IBM XL C\C++ version 13.1.5) with the activated support of the parallel execution in the frame of IBM spectrum Message Passing Interface (MPI) version 10.1.0.

Calculations have been performed with the use of 1–3 computational nodes (8–60 cores, 1–6 co-processors respectively) where each MPI-process corresponds to one processor core. Their binding to co-processors was performed by the setup of the CUDA_VISIBLE_DEVICES environment variable so that MPI processes running on the first processor cores would be linked to the first co-processor, and processes of the second processor cores would be linked to the second co-processor. Calculations were terminated after performing the full self-consistent loop for each case.

The CPU machine has been used for the comparison is the Cray XC30 (Cray Inc. 901 Fifth Avenue, Suite 1000 Seattle, WA, USA) where each compute-node contains two Intel 2.7 GHz 12-core E5-2697 v2 (Ivy Bridge) series processors with the supported two hardware threads per core. The two processors are connected by two QuickPath Interconnect (Intel Corporation 2200 Mission College Blvd. Santa Clara, CA, USA) links with local memory of 32 GB. For the sake of consistency in the second CrI${}_{3}$ case, the same Quantum ESPRESSO 6.2 version was used coherently both on the Cray and POWER8 machines, while the first system proceeded using version 5.4.

## 3. Computational Details

The ground states calculations were performed using a computational implementation of the density-functional theory (DFT) package Quantum ESPRESSO [11]. We used an ultrasoft (US) pseudopotential approach along with the PAW method and the Perdew-Burke-Ernzerhof generalized gradient approximation [12] of the exchange and correlation functional within its spin-polarized version. For scalar relativistic and relativistic pseudopotentials, the 600 Ryd charge density cut-off and 60 eV wave function cut-off was employed to optimize the ground state. The 18×18×1 (for a double layer material) and 7×7×1 (for the single layer material) Monkhorst and Pack *k*-point meshes [13] were used for integration in the irreducible Brillouin zone by a special-points technique with broadening σ=0.02 Ry according to the Marzari-Vanderbilt cold smearing method [14]. Thus, these meshes ensure convergence of total energy to less than $10^{-6}$ eV/atom. By processing the output data from Quantum ESPRESSO, the Hamilton-weighted populations were calculated based on periodic plane-wave DFT output with the aid of the ‘LOBSTER’ software package  [15]. For the treatment of multi-layered systems, Van der Waals corrections were used based on the semiempirical Grimme’s DFT-D2 approach.

It is well-known that for *d*-electrons in chromium compounds one has to take into account correlation effects in order to describe their electronic and magnetic properties correctly [16,17]. A density functional theory plus Hubbard U method is the most efficient and widely used approach to investigate strongly correlated semiconductors [18]. In this paper, for the main GGA+U calculations in the frame of DFT, we employed the simplified scheme introduced by Dudarev et al. [19]. We used $U_{e}ff$ values which have been computed previously using a linear response method [20]—for 2D CrI${}_{3}$, CrBr${}_{3}$ and CrCl${}_{3}$
$U_{e}ff$ are equal to 2.65 eV, 2.68 eV, and 2.63 eV, respectively. For the treatment of 5d/4d electron correlation effects in the Ru-based compounds, we used the recommended value 1.6 eV [21] for $U_{e}ff$.

## 4. Computational Performance

In the first instance, we performed a standard QE PSIWAT test [11], which allowed us to identify the best-performed build version of the code. The corresponding test system is a gold surface covered by thiols in interaction with water (see Figure 1). Three versions of the software builds have been tested—CPU, GPU, and ESSLCUDA. Benchmarks result for the computational acceleration *S* as a function of processors number *N* are presented on the Figure. Where the three abovementioned builds of the code were employed. The acceleration has been computed as follow $S=T_{8}/T_{N}$, where $T_{8}$ is the average wall time using 8 cores for the first build version of the code, and $T_{N}$—is an average wall time of code in the consideration. Looking onto Figure 1, it is clear that for any number of MPI processors, GPU and ESSLCUDA builds pose the best performance over the co-processor disable version. Furthermore, with the use of the GPU-enabled version, it was possible to performer calculations about 3.3 times faster, where, in addition to eight cores, one co-processor was involved. The best performance was observed for the ESSLCUDA build of the package. The minimal wall time reached is 553 s, which is 2.1 and 1.6 times faster than testing results for the CPU and GPU build on the same cluster. In the interest of understanding the influence of k-point parallelization onto the performance, we also performed calculations with the use of only one k-points pool (npool = 1). This clearly indicates the performance degradation against the k-grid distribution over 4-nodes (npool = 4), where computational enhancements can be achieved by increasing the number of MPI-processors in use.

The next step was with the use of the best performed QE build to compare its performance with other nation-wide top-ranking supercomputer facilities. The Cray XC30 MPP supercomputer equipped with a high-performance Lustre storage system was chosen as its counterpart. To emphasize the role of the unit cell size and k-points mesh for a purer 2D system, there are two test cases that have been considered.

Case 1 is the mono-layered system containing 4 Cr atoms and 12 I atoms with a relatively dense k-points mesh 18 × 18 × 1 (164 k-points in total) and extremely large vacuum space 43 Å where PAW pseudopotentials have been used. The second case is a single layer of CrI${}_{3}$ (25 k-points in total) where smaller vacuum space 25 Å and loose 7 × 7 × 1 k-points mesh have been applied along with US pseudopotentials. The aim here was to make the system computationally heavier not only by including more atoms but also by utilizing a more dense grid. This should reveal how efficient the k-points parallelization implemented into the QE package is.

The results of the scaling analysis are presented in Figure 2. This graph is quite revealing in several ways. First, unlike the reasonable expectations, it is clear that the scalability for Case 2 reaches the limitation. Case 2 scaling was better and related to the presence of a relativity larger number of k-points, where each k-point pool can process separately on each computational core. Secondly, performing polynomial fitting of the results, we identified that a quadratic term ($\propto 1/N^{2}$) significantly contributed to T(N) trends. However, this can be explained as a processing slowing due to the inter-node connection where N × N operations need to be performed. Furthermore, it can be resolved by using the -npool option for k-points distribution over separate nodes. At the same time, what can be clearly seen in this figure is the constant dominance of the hybrid system performance. Obtained results for a double-layered system of CrI${}_{3}$ magnet presented on Figure 3.

## 5. Accelerating with GPU Computing Example—FM-AFM Exchanges Competition in 2D Magnets

To further examine electronic structure of CrX${}_{3}$ compounds we utilized a specific bonds characteristic—the electronic localization function $\mathrm{ELF}={\left(1+\chi_{\sigma }^{2}\right)}^{-1}$ (where χ is a dimensionless localization index relativity to the uniform-density electron gas). It can portray the localization nature of electrons in the system [22]. It has been calculated for all of the CrX${}_{3}$ materials and obviously indicates their ionic character, where electrons have been localized around halides forming Cr-X ionic bonds. Likewise, the valency of Cr atoms has been computed using the Bader analysis method [23]. The increase of Cr ion valency along with the covalency of p−d bonds was observed by going from *I* with the biggest ionic radius up to the *F*, where the ferromagnetic exchange gets weaker and is less preferable than the anti-ferromagnetic (see Figure 4).

Results of ELF calculations for the CrX${}_{3}$ materials shown on the Figure 4 obviously indicate the presence of ionic character, where electrons localized around halides forming Cr-X bonds. However, we found that the ionicity of Cr atoms tends to decrease going from the I up to F halogen by bringing more covalency to Cr-X bonds and widening the band gap. Looking to computed charge density maps and projected crystal orbital Hamilton populations (pCOHP), one can clearly see covalent bridge formation due to the strengthening of Cr-X bonds. Those facts are consistent with a ligand-field theory [24], where assuming constant on-cation Hubbard repulsion $U_{3d}$ for a bigger $t_{2g}-e_{g}$ orbitals separation $\Delta_{t_{2g}-e_{g}}$, one can expect the FM exchange to be less preferable against the AFM contribution $J^{AFM}\approx 0.25\Delta_{t_{2g}-e_{g}}^{2}/U_{3d}$ [25]. However, this kind of picture differs from the competition mechanism between ferromagnetic superexchange mediated through the X ions and direct antiferromagnetic exchanges in well-studied metal halides MX${}_{2}$ [26,27,28]. Therefore, the deeper understanding of the p−d hybridization in the 2D Cr$X_{3}$ halides is the key point for the correct description of microscopic mechanisms where indirect super-exchange will play a major role.

For the Hubbard-like model describing magnetism of localized magnetic moments $\vec{S_{i}}$ of *d*-shells it can be shown that the second order perturbative expansion in the large Coulomb interaction limit $U_{dd}>>t_{eff}$ yields the standard Heisenberg picture with a well know anti-ferromagnetic exchange value $J^{AFM}=-2t_{eff}^{2}/U_{dd}$. However, this kind of simplified picture is not suitable for a correct description of magnetic transition metal (TM) compounds where the metal atom-metal atom magnetic exchange is mediated by ligands. Therefore, even for a simple model of the metallic atoms dimer, the effect of charge transfer should be considered [29] where the total exchange $J\approx 2l^{\prime }-4{\left(t_{eff}+l^{\prime }\right)}^{2}/U_{dd}$ possesses a competitive character between FM and AFM exchanges due to the two-electron ionic integral $l^{\prime }$. Moreover, in the recent study [30], the importance of ligand-field phenomena and charge-transfer transitions for the optoelectronic properties of 1L CrI${}_{3}$ demonstrated. Therefore, the bigger increase of energetic cost for the charge transfer $l^{\prime }$ in comparison to the $t_{eff}$ explains the reduction of FM exchange contribution in the MX${}_{3}$ halides: for the large halide ions, it is easy to donate electrons and form more covalent bonds. So by moving X in the 17th column of the periodic table from the Br with a higher ionicity up to the F with lesser ionicity, the ferromagnetic impact become weaker in the comparison to anti-ferromagnetic one. In the case of Cr$X_{3}$, it leads to the situation where the antiferromagnetic order become more preferable for the CrF${}_{3}$, in contrast to other ferromagnetic compounds of that family. On the macroscopic level, those processes can be reformulated as follows: by the band gap increase the $p-e_{g}$ coupling gets smaller since it is harder for electrons to move from *p* to the $e_{g}$-orbitals. Hence, the $t_{2g}-p-{t_{2g}}^{\prime }$ electron path corresponding to the AFM exchange become more preferable, while transfer along $t_{2g}-p-e_{g}^{\prime }$ and $e_{g}-p-e_{g}^{\prime }$ paths are impeded, which correspond to the FM exchange between *d* elements.

However, that simplified inclusion of charge-transfer effects into the physical model is not good enough for a proper quantitative description of magnetic phenomena in bulk TM materials, where the interplay between AFM and FM contributions is important. In a more sophisticated manner, these ratios for the M${}^{+3}$ sublattice in the first nearest-neighbour 〈i,j〉 approximation can be described as [31]:(1)$$\begin{array}{c} H=-\sum_{\langle i,j\rangle }J^{FM}\left(1-F\right)\left({\vec{S}}_{i}\cdot {\vec{S}}_{j}\right); \end{array}$$(2)$$\begin{array}{c} F=|\frac{J^{AFM}}{J^{FM}}|=\frac{t_{t_{2g}-{t_{2g}}^{\prime }}^{2}}{t_{t_{2g}-{e_{2g}}^{\prime }}^{2}}\cdot \left(\;\frac{1+{\tilde{\Delta }}^{2}}{{\tilde{J}}_{H}}-1-\tilde{\Delta }\right), \end{array}$$
where quantities with a tilde—the bandgap $\tilde{\Delta }=\Delta /U$ and Hund’s coupling ${\tilde{J}}_{H}=J_{H}/U$, are normalized to the Hubbard repulsion *U*.

By keeping a metallic ion M unchanged in MX${}_{3}$ compounds *U* and $J_{H}$, they can be assumed as slow-changing quantities due to their local character depending on *d*-orbitals. Hence, by substituting halide atoms X, one can control AFM/FM contributions to the total exchange due to $t_{t_{2g}-{t_{2g}}^{\prime }}$, $t_{t_{2g}-{e_{2g}}^{\prime }}$ and Δ parameters change. Reaching the situation where F≈1 it is also important to consider the effects of antisymmetric exchange interactions beyond the first coordination sphere along with the higher-order $t^{2}/U$ terms like a bi-quadratic (BQ) $B\cdot {\left({\vec{S}}_{i}\cdot {\vec{S}}_{j}\right)}^{2}$ and 4-spins interactions [32]. Since for 2D magnets, BQ exchange $B\approx -2\left(t_{t_{2g}-{t_{2g}}^{\prime }}^{4}+t_{t_{2g}-{e_{2g}}^{\prime }}^{4}-4t_{t_{2g}-{t_{2g}}^{\prime }}^{2}\cdot t_{t_{2g}-{e_{2g}}^{\prime }}^{2}\right)/{\left(U+J_{H}\right)}^{3}$ [33] can be a comparable quantity with a bilinear (BL) exchage [34] the proper account of such effects can lead to the easy-magnetization direction switches and is, thereby, crucial for correct magnetic states descriptions [35,36]. The noticeable impact of the bi-quadratic exchange onto the optical absorption spectra have been observed even for a bulk ferromagnetic system, like ruby, where Cr${}^{+3}$ ions incorporated into octahedral positions [37].

In order to demonstrate the importance of higher-order exchanges, we performed GPU-enhanced non-collinear calculations for rotation spins in the RuCl${}_{3}$ compound. To extract high-order exchange we used techniques proposed by Kartsev et al. [38]. It is clearly seen from Figure 5 that the higher-order effects are in evidence, and the inclusion of BQ exchange to the model leads to a more accurate description of spin subsystem behaviour. This can be explained as an admixture effect of two pseudo-spin states $J_{eff}=1/2$ and $J_{eff}=3/2$ [39]. While for $S>\frac{1}{2}$ the BQ term ${\left(S_{i}\cdot S_{j}\right)}^{2}$ can not be reduced into a form of the usual bi-linear Heisenberg exchange $\left(S_{i}\cdot S_{j}\right)$ and, hence, is nonnegligible. The $E^{tot}\left(\theta \right)$ 30 points data have been obtained almost three times faster using GPU-aided code in comparison to code employing CPU only.

## 6. Conclusions

In summary, the results observed indicate the efficiency of hybrid computer clusters for collinear and noncollinear first-principle calculations where we employed the IBM ESSL mathematical library with automatic partial offloading of calculations into co-processors. This approach allowed us to increase the computational speed of the application based on the BLAS library, which originally was not designed for use on graphical processors. The proposed setup of a computing environment has been effectively applied for computing the magnetic and electronic characteristics of 2D magnets demonstrating the peculiar ferro- and antiferromagnetic exchange interplay and related high-order effects. Therefore, the proposed methodology will speed up computational schemes for low-dimensional materials design using the premier GPU-accelerated platforms. Accordingly, it paves the way to improve the performance of the accelerated materials design based on high-throughput computational schemes.

## Figures and Tables

**Figure 1 nanomaterials-11-02967-f001:**
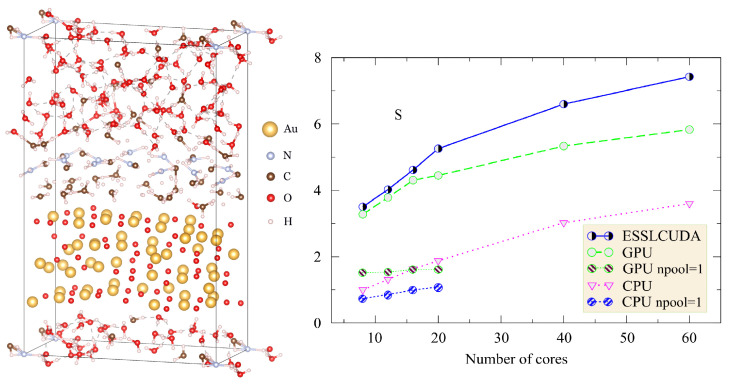
The orthorhombic unit cell used for simulating moistening of the gold surface coated with thiol. The second panel shows computational acceleration as a function of core number for different assemblies used for that cases calculations.

**Figure 2 nanomaterials-11-02967-f002:**
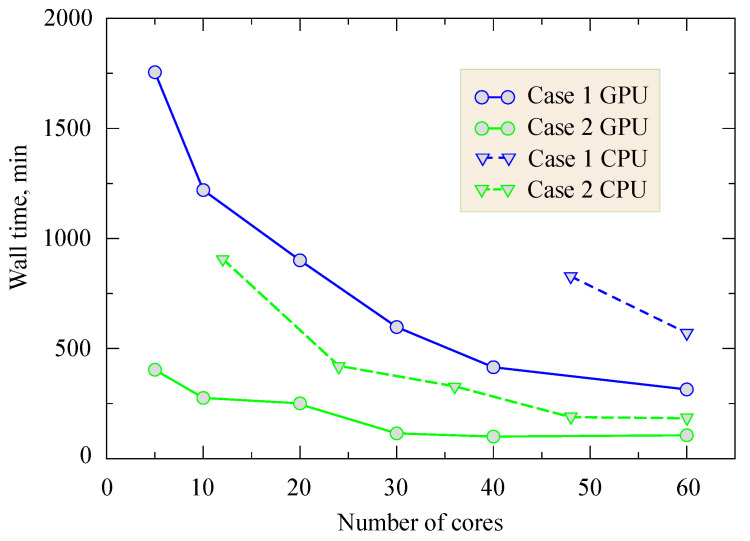
Wall time as a function of core number for different cases used for test calculations. Computational acceleration as a function of MPI core number. Case 1 (hard case) is the monolayer (4 Cr and 12 I atoms) k-points mesh 18 × 18 × 1 (164 k-points in total) and vacuum space 43 Å. Case 2 (mild case) is the single layer of CrI${}_{3}$ (25 k-points in total) where smaller vacuum space 25 Å and loosened 7 × 7 × 1 k-points mesh have been applied along with US pseudopotentials.

**Figure 3 nanomaterials-11-02967-f003:**
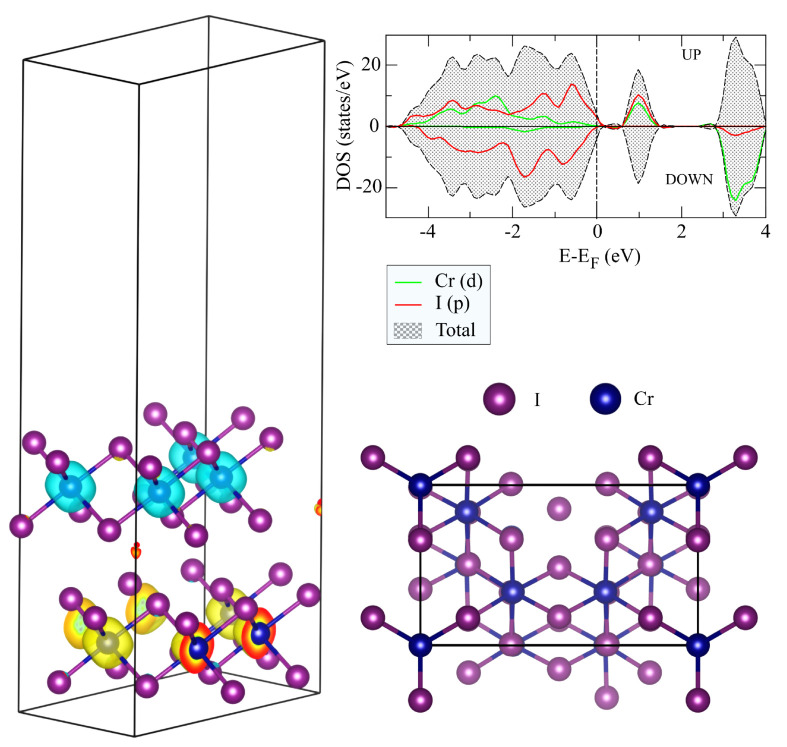
Orthorhombic unit cell used for simulating two CrI${}_{3}$ layers coupled antiferromganetically and isosurface of the magnetization density (light green for spin-up and yellow for spin-down channels correspondingly). The top view of the unit cell. Computed total and partial density of states.

**Figure 4 nanomaterials-11-02967-f004:**
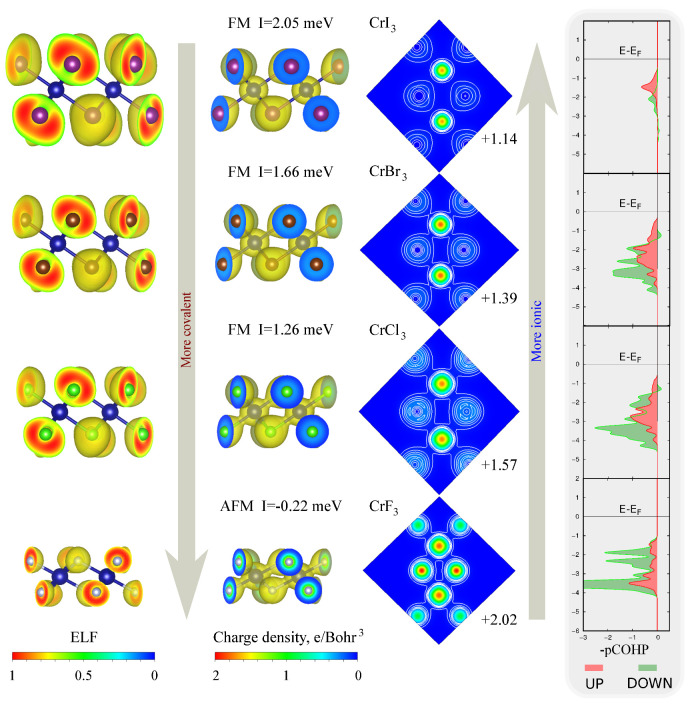
3D plots are charge density maps and distributions of the electronic localization function in the unit cell of CrX${}_{3}$ compounds with 0.5 and 0.05 e/Bohr${}^{3}$ isosurfaces. The 2D cuts are charge density maps in the (Cr-X-Cr) plane with 0.05 e/Bohr${}^{3}$ isolines (0–0.3 e/Bohr${}^{3}$) where the left and right arrows indicate the ionicity and covalency change correspondently. Numbers near the right and left arrows correspond to the Bader charges of Cr atoms and total magnetic exchange $J^{FM}+J^{AFM}$. pCOHP graphs for up and down spin channels placed into grey inserts indicate the strength of orbital overlaps for Cr−X bonds where the energy axis is shown relative to the Fermi level.

**Figure 5 nanomaterials-11-02967-f005:**
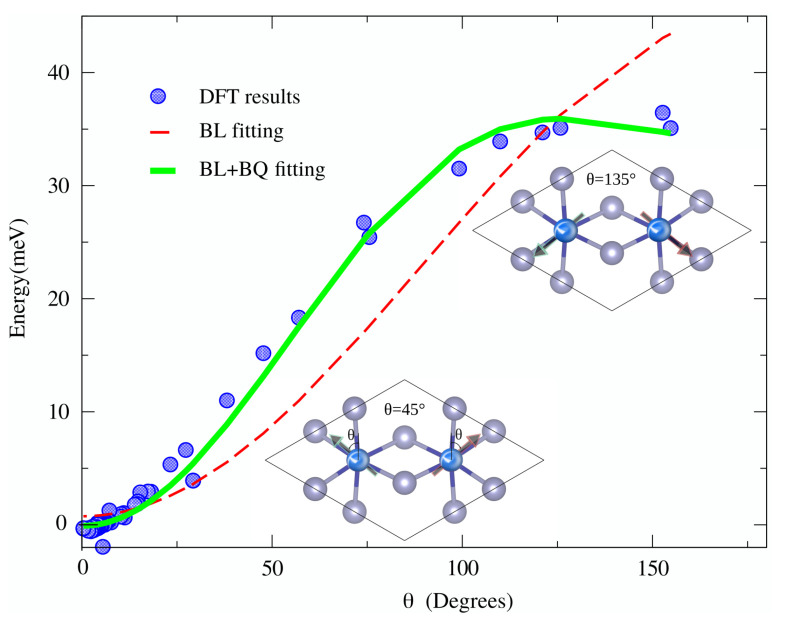
The presence of the higher-order exchange can be capture by performing constrained non-collinear DFT calculations where magnetic moments rotate inplane around the principal axes. Calculated total energy $E^{tot}$ as a function of θ (dots) and different fits using the bilinear Heisenberg and the biquadratic Heisenberg model. The insets show the definition of the spins rotating by angle θ relative to the vertical axis in a unit-cell plane of RuCl${}_{3}$ and spin configurations at θ = 45${}^{\circ }$ and 135${}^{\circ }$.

## Data Availability

Not applicable.

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
