# Peer review of "Analysis of Ionicity-Magnetism Competition in 2D-MX3 Halides towards a Low-Dimensional Materials Study Based on GPU-Enabled Computational Systems"

_nanomaterials, 2021, doi:10.3390/nano11112967_

Round 1

Reviewer 1 Report

The work deals with GPU enhanced calculations of selected solid state compounds with the Quantum Espresso package.

After selecting the best computational scheme - ESSLCUDA - authors proceeded to the calculations of CrX3 and RuCl3 compounds. Author likewise performed analysis of the electronic structure of CrX3 using available tools (ELF, pCOHP etc).

The referee has few minor comments:

In l.149 is the mention of  acronyms FM-AFM - they should be explained in the text before. The ELF, pCOHP and others acronyms are explained and referenced as expected.

In l.201 there is the TM acronym, please explain that also in the text.

Other typos:

l.119 "ESSLCUDA" instead of "ESSSLCUDA"

l.150 "To further examine" instead of "To further exam"

l.153 "It has been calculated" instead of "It been calculated"

Reviewer 2 Report

Over the last two decades, the general purpose GPU technology has been rapidly developed, the computational performance of GPUs has been increasing much more quickly than that of conventional CPUs. And the GPUs have been used in the computational physics, especially in the fields of large-scale classical molecular dynamics and the first-principle calculation of materials.  The extensive exploring of GPU v.s. CPU calculation performance is necessary.

The present work tends to explore the scalability and performance of Quantum Espresso package in the hybrid CPU-GPU environment. The CPU-GPU results are helpful for further GPU code development and its applications. However, I don’t recommend the acceptance of the manuscript in the present form. The manuscript contains three parts of studies. They are (1) section-1,2,3,4, the study of CPU-GPU performance; (2) the first 5 paragraphs of section-5, the macroscopic (I believe it should be microscopic) origin of FM-AFM exchanges in CrX3 (‘exchange’ without ‘s’ should be better) competition; (3) the last paragraph of sectin-5, high-order exchanges in RuCl3. They are three isolated studies, they do not have much logical relevance. It is strange to put these three isolated studies into one manuscript. I strongly suggest the authors to reorganize the manuscript.

   Besides the manuscript organization problem, I have the following further questions:

(1)         The cut-off energy for wavefunction is only 60 eV (see line-85), which is unreasonably low in DFT calculations. Please test this parameter.

(2)         The k-point meshes for double-layer and single-layer materials are 18x18x1 and 7x7x1, respectively (see line-86,87). However, the in-plane cell vector lengths for double-layer and single-layer materials are almost the same. So the in-plane k-mesh should be the same. Why did the authors choose much more dense k-mesh for double-layer material?

(3)         The ‘case 1’ and ‘case 2’ in Figure. 2 and the corresponding caption are ambiguous, and there is no much explanation about the two cases. The authors should make the definitions clear both in the figure and main text.

Author Response

Please have a look at the attached pdf-file

Round 2

Reviewer 2 Report

The authors have answered all of my questions and the paper has been greatly improved. Therefore, it can be accepted for publication.